# Oculo-Cutaneous Albinism Type 4 (OCA4): Phenotype-Genotype Correlation

**DOI:** 10.3390/genes13122198

**Published:** 2022-11-23

**Authors:** Ester Moreno-Artero, Fanny Morice-Picard, Eulalie Lasseaux, Matthieu P. Robert, Valentine Coste, Vincent Michaud, Stéphanie Leclerc-Mercier, Dominique Bremond-Gignac, Benoit Arveiler, Smail Hadj-Rabia

**Affiliations:** 1Department of Dermatology, and Reference Centre for Genodermatoses and Rare Skin Diseases (MAGEC), Université Paris Descartes-Paris Cité, INSERM U1163, Institut Imagine, APHP, Hôpital Universitaire Necker-Enfants Malades, 75015 Paris, France; 2Department of Dermatology, Reference Centre for Genodermatoses and Rare Skin Diseases, INSERM U1312, Bordeaux University Hospital, 33000 Bordeaux, France; 3Department of Medical Genetics, Bordeaux University Hospital, 33000 Bordeaux, France; 4Department of Ophthalmology, and Reference Center for Rare Eye Diseases (OPHTARA), Université Paris Cité, Hôpital Universitaire Necker-Enfants Malades, APHP, 75015 Paris, France; 5Borelli Centre, UMR 9010, CNRS-SSA-ENS Paris Saclay-Université Paris Cité, 91190 Paris, France; 6Department of Ophthalmology, Bordeaux University Hospital, 33000 Bordeaux, France; 7INSERM U1211, Laboratory for Rare Diseases, Genetics and Metabolism, 33000 Bordeaux, France; 8Department of Pathology, and Reference Centre for Genodermatoses and Rare Skin Diseases (MAGEC), Université Paris Descartes-Paris Cité, APHP, Hôpital Universitaire Necker-Enfants Malades, 75015 Paris, France; 9INSERM, UMRS 1138, Team 17, From Physiopathology of Ocular Diseases to Clinical Development, Université Paris Descartes-Paris Cité, 75015 Paris, France

**Keywords:** oculo-cutaneous albinism, OCA4, *SLC45A2*, hypopigmentation, genetics, foveal hypoplasia

## Abstract

Albinism is a genetic disorder, present worldwide, caused by mutations in genes affecting melanin production or transport in the skin, hair and eyes. To date, mutations in at least 20 different genes have been identified. Oculo-cutaneous Albinism type IV (OCA4) is the most frequent form in Asia but has been reported in all populations, including Europeans. Little is known about the genotype-phenotype correlation. We identified two main phenotypes via the analysis of 30 OCA4 patients with a molecularly proven diagnosis. The first, found in 20 patients, is clinically indistinguishable from the classical OCA1 phenotype. The genotype-to-phenotype correlation suggests that this phenotype is associated with homozygous or compound heterozygous nonsense or deletion variants with frameshift leading to translation interruption in the *SLC45A2* gene. The second phenotype, found in 10 patients, is characterized by very mild hypopigmentation of the hair (light brown or even dark hair) and skin that is similar to the general population. In this group, visual acuity is variable, but it can be subnormal, foveal hypoplasia can be low grade or even normal, and nystagmus may be lacking. These mild to moderate phenotypes are associated with at least one missense mutation in *SLC45A2*.

## 1. Introduction

Albinism is an autosomal recessive or X-linked disorder, present worldwide, that affects melanin production in the skin, hair, and eyes [1]. Variants in 20 genes are responsible for variable hair and skin pigmentation impairment and visual alterations [2]. Recently, an additional gene, *PMEL*, has been suggested to cause oculo-cutaneous albinism [3]. Visual anomalies include foveal hypoplasia, hypopigmentation of the retina, and abnormal decussation of the ganglion nerve fibers at the optic chiasm, resulting in nystagmus, low vision, and photophobia [4,5]. Patients with albinism have an increased risk of developing non-melanoma skin cancers due to the defect of melanin synthesis. Daily photoprotection of the skin and eyes, beginning in childhood, is therefore essential.

Oculo-cutaneous albinism type IV (OCA4) is caused by bi-allelic variants in the *SLC45A2* gene [6]. The encoded protein, membrane-associated transporter protein (MATP), is involved in melanosome pH regulation and protein transport [6]. While it is the most frequent form of albinism in Asia, OCA4 accounts for only about 10% of albinism cases globally [7].

The broad spectrum of OCA4 phenotypes varies from pale skin, a complete absence of hair pigment, pink nevi, and blue eyes associated with classic ocular abnormalities, to mild hair hypopigmentation (light brown, or even black, hair), and a skin color comparable to the general population [7].

Here we detail the clinical characterization of 30 OCA4 patients from 26 families aiming to find a genotype-phenotype correlation.

## 2. Methods

Patients with OCA4 were retrospectively included. The diagnosis of OCA4 was confirmed when molecular analyses identified pathogenic variants in *SLC45A2* by next-generation sequencing (NGS) using a panel of genes involved in syndromic and non-syndromic albinism (*TYR*, *OCA2*, *TYRP1*, *SLC45A2*, *SLC24A5*, *C10orf11*, *DCT*, *GPR143*, *HPS1* to *HPS11*, *CHS1*). Genetic, dermatological, and ophthalmological data were collected. Informed consent was obtained from each patient or their legal representatives when minor. The local ethics committee of the University Hospital of Bordeaux and the University Necker-Enfants Malades Hospital, France, approved the study.

## 3. Results

Between 2014 and 2021, 30 patients [13 males, 17 females, mean age = 23.2 years, median age = 30 years (2–66)] belonging to 26 unrelated families from France (25 patients), Morocco (3 patients, P1–P3), China (1 patient, P22) and Mauritania (1 patient, P29) were included (Table 1). Five patients from 4 families were born from a consanguineous union. The following patients belong to 4 different sibships: P2 and P3; P6 and P7; P10 and P11; P26 and P27.

### 3.1. Dermatological Phenotype

Hair (scalp and body) ranged from completely white (18 patients) to completely blond (8 patients, Table 1). The 4 remaining patients had white-blond (2 patients), red-blond (1 patient) or dark blond hair (1 patient). Eyelashes and eyebrows were white (20 patients), blond (9 patients) or brown (1 patient). Hair, eyelashes, and eyebrows had the same color in 26 patients, while eyelashes and eyebrows were clearer in the 4 remaining patients. 

Mild cutaneous hypopigmentation was observed in 10 patients (P21–P30), while the other 20 patients presented with clearly hypopigmented skin (phototype I). All patients presented with a clearer phototype, including hair, eyelashes and eyebrows, than the phototype of each of their parents.

Twenty-two patients presented with multiple nevi that were amelanotic in 10 and pigmented in 12 patients. Six patients experienced sunburns and solar lentigines associated with a poor level of photoprotection, while adapted photoprotection was reported in 15 patients. There was no skin cancer.

### 3.2. Ocular Phenotype

All but 3 patients had infantile nystagmus syndrome. Eye color ranged from blue (19 patients) to brown (2 patients). Nine patients had blue-grey iris. Mean binocular visual acuity was 2/10, ranging from 0.5/10 to 9/10. Only one patient (P2) exhibited severe visual impairment: visual acuity < 1/10 according to the International Classification of Disease (ICD) 11. Foveal hypoplasia could be graded in 26 patients: grade 4 (17 patients), grade 3 (6 patients), grade 2 (2 patients) and grade 1 (1 patient). Patient P22 had a visual acuity of 9/10 in the right eye and 7/10 in the left and a grade 1 foveal hypoplasia. Patient 30 had a binocular visual acuity of 5/10. The latter two patients had a mild dermatological phenotype (see above). The ocular phenotype of patients is displayed in Table 1.

### 3.3. Molecular Findings

The variants are reported in Table 1. Five novel variants were identified: c.267_271del; c.258del; c.533_534dup; c.977T>A and c.1255C>A. Ten patients (33.33%) carried two deletions in either the homozygous (n = 5, P1–P3, P9 and P18) or compound heterozygous state (n = 5, P10–P13 and P20). Twelve patients (40%) were compound heterozygous carriers of a deletion and a missense variant (P4–P8, P15, P17, P19, P23, P24, P26 and P27). Six patients (20%) carried two missense variants in either the homozygous (n = 2, P16 and P29) or compound heterozygous state (n = 4, P22, P25, P28 and P30). The last two, P14 and P21, were homozygous for a nonsense variant and compound heterozygous for duplication and a missense variant, respectively. The p.(Thr329Lysfs*69) variant was identified in 12 patients (40%, P5–P11, P15 and P17–P19).

## 4. Discussion

From an analysis of 30 OCA4 patients with a molecularly proven diagnosis, we identified two main phenotypes. The first phenotype, found in 20 patients (group 1 = P1–P20), is clinically indistinguishable from the classical OCA1 phenotype (Figure 1A,B) [5]. The genotype-to-phenotype correlation suggests that this phenotype is associated with homozygous or compound heterozygous nonsense or deletion variants with frameshift leading to translation interruption in the *SLC45A2* gene. The second phenotype, found in 10 patients (group 2 = P21–P30), is characterized by very mild hypopigmentation of hair (light brown or even dark hair) and skin that is similar to the general population (Figure 1C). In this group, visual acuity is variable, but it can be subnormal, foveal hypoplasia can be low grade or even normal, and nystagmus may be lacking. These mild to moderate phenotypes are associated with at least one missense mutation in *SLC45A2*.

All 20 patients of the first group presented with white-blond or white platinum hair, major skin hypopigmentation, blue eyes, and moderate to severe ophthalmologic features (Table 1). Among them, 6-year-old twins (P2 and P3) presented with severe ocular and dermatological phenotypes, as well as neurological manifestations. The sister (P2) had a slight psychomotor development delay, probably related to her impaired vision. The boy (P3) had an overall delay of acquisitions and a pervasive developmental disorder.

A moderate dermatological phenotype, reported in 10 patients (group 2), is characterized by blond, dark blond or red-blond hair, and blue or brown iris in P22 and P29. Skin color was similar to the general population in Europe. Skin and hair were hypopigmented in patients P22 and P29 when compared to their relatives and population background (China and Mauritania, respectively). With the exception of patient P22 (visual acuity of 9/10 in the right eye and 7/10 in the left, and a grade I foveal hypoplasia) and P30 (binocular visual acuity of 5/10), mild to moderate visual impairment was reported in the remaining 8 patients (Table 1).

In our series, carrying two homozygous or compound heterozygous deletions is associated with the most severe phenotype. Among patients of group 1, 90% carried two deletions (either homozygous or compound heterozygous) or were compound heterozygous for a deletion and a missense variant. The last 2 patients (P14 and P16) presented with a homozygous nonsense and a homozygous missense variant, respectively. Similarly to Rundshagen et al. in their German series, a high prevalence (40%) of the p.(Thr329Lysfs*69) variant was found here. In both series, the severity of the associated phenotype is very similar: very pale skin, white-yellowish hair with little or no further pigmentation during life, and a visual acuity between 20/400 and 20/200 [8]. Unfortunately, the phenotype associated with p.(Lys389Serfs*55) and p.(Gly491Arg) variants is lacking [9,10,11].

A moderate dermatological phenotype is associated neither with the presence of two deletions, nor with the presence of the p.(Thr329Lysfs*69) variant. The phenotypes associated with p.(Tyr317Cys) and (p.(Ala511Val) variants were reported as dark blond hair and a lesser degree of ocular hypopigmentation, without marked visual impairment, and dark blond or golden hair, light brown skin color and bluish eyes, respectively [8,12]. The synonymous variant p.(Val506=), in association with another allelic variant, could allow for the synthesis of a residual amount of functional protein and explain moderate hypopigmentation reported in patients P24 and P28.

Patients with major skin and hair hypopigmentation had moderate visual impairment, except P2, who presented with severe visual acuity impairment, according to the International Classification of Disease (ICD) 11. Patients with very mild hypopigmentation of hair and skin presented with subnormal visual acuity (P22 and P30), up to a mild or moderate degree of visual acuity impairment. In the literature, three OCA4 patients with severe hypopigmentation, including iris transillumination, normal visual acuity, foveal development, and visual nerve routing, have been described. Two were compound heterozygous for a missense variant, while the third carried a missense heterozygous variant associated with a splicing variation [13]. Thus, the absence of melanin does not systematically lead to foveal hypoplasia and visual misrouting.

Finally, we suggest that depending on the genetic background, the impact of loss-of-function of *SLC45A2* is different, leading to either a mild or a classical OCA1-like phenotype. Moreover, in our series, it seems that carrying the *SCL45A2* p.(Thr329Lysfs*69) variant in either the homozygous or compound heterozygous state is associated with a high risk of having a severe phenotype. Our findings warrant further studies to understand the phenotypic variability of OCA4.

## Figures and Tables

**Figure 1 genes-13-02198-f001:**
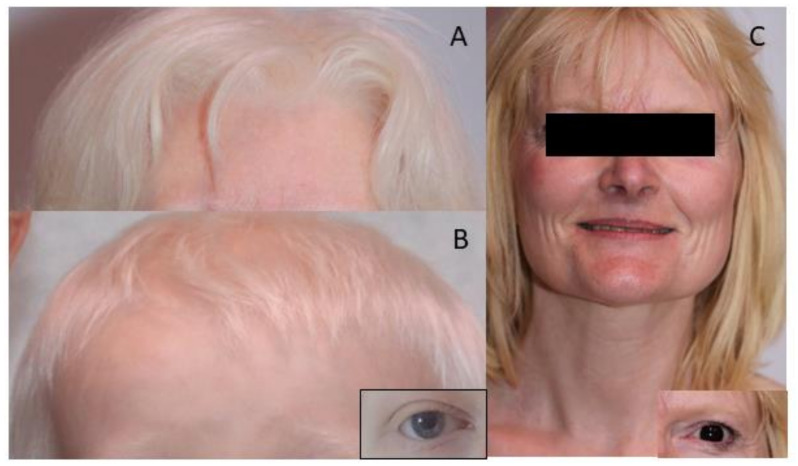
The clinical picture of patients from group 1 and group 2. Patients P17 (**A**) and P7 (**B**) belong to group 1. They presented with white hair, eyelashes and eyebrows. Patient P29 (**C**, group 2) presented with blond hair and brown eyes.

**Table 1 genes-13-02198-t001:** Clinical manifestations and *SLC45A2* identified variants in 30 patients.

Patients	Gender, Age (Years)	Genetic Background	Consanguinity	Nevi	Eyes	Hair	Eyebrows Eyelashes	Nystagmus	Strabismus	VA	Refraction	ITI	MT	FHP	Variant 1 (SLC45A2 NM_016180.5)	Variant 2 (SLC45A2 NM_016180.5)
**Group I**																
P1	M, 20	Morocco	Yes	Present, amelanotic	Blue	White	White	Yes	Yes, esotropia	1.6/10 RE; 2/10 LE	Hypermetropia astigmatism	NA	NA	Grade IV	**NM_016180.5(SLC45A2):c.267_271del** **Chr5(GRCh37):g.33984422_33984426del** **p.(Ser90Glnfs*42)**	**NM_016180.5(SLC45A2):c.267_271del** **Chr5(GRCh37):g.33984422_33984426del** **p.(Ser90Glnfs*42)**
P2	F, 7	Morocco	Yes	Present, pigmented	Blue	White blond	White	Yes	Yes, left exotropia	1/20 RE; 1/20 LE	Hypermetropia Astigmatism	Grade IV	Grade II	Grade IV	NM_016180.5(SLC45A2):c.1028_1029del Chr5(GRCh37):g.33954469_33954470del p.(Gly343Alafs*10)	NM_016180.5(SLC45A2):c.1028_1029del Chr5(GRCh37):g.33954469_33954470del p.(Gly343Alafs*10)
P3	M, 7	Morocco	Yes	Present, pigmented	Blue	White blond	White	Yes	No	NA	Hypermetropia Astigmatism	Grade III	Grade II	Grade IV	NM_016180.5(SLC45A2):c.1028_1029del Chr5(GRCh37):g.33954469_33954470del p.(Gly343Alafs*10)	NM_016180.5(SLC45A2):c.1028_1029del Chr5(GRCh37):g.33954469_33954470del p.(Gly343Alafs*10)
P4	F, 49	France	No	Absent	Blue grey	White	White	Yes	No	2/10 RE; 2/10 LE	NA	Grade IV	NA	Grade IV	NM_016180.5(SLC45A2):c.273del Chr5(GRCh37):g.33984417del p.(Ser92Alafs*21)	NM_016180.5(SLC45A2):c.1068C>G Chr5(GRCh37):g.33951747G>C p.(Asn356Lys) (N356K)
P5	F, 63	France	No	Present, amelanotic	Blue	White	White	Yes	No	2/10 RE; 3/10 LE	NA	Grade III	NA	Grade III	NM_016180.5(SLC45A2):c.986del Chr5(GRCh37):g.33954512del p.(Thr329Lysfs*69)	NM_016180.5(SLC45A2):c.1036G>T Chr5(GRCh37):g.33951779C>A p.(Val346Leu) (V346L)
P6	M, 18	France	No	Present, pigmented	Blue	White	White	Yes	No	1/10 RE; 1/10 LE	NA	Grade IV	NA	Grade IV	NM_016180.5(SLC45A2):c.986del Chr5(GRCh37):g.33954512del p.(Thr329Lysfs*69)	NM_016180.5(SLC45A2):c.1471G>A Chr5(GRCh37):g.33944875C>T p.(Gly491Arg) (G491R)
P7	M, 9	France	No	Absent	Blue	White	White	Yes	No	1/10 RE; 1/10 LE	NA	Grade IV	NA	Grade IV	NM_016180.5(SLC45A2):c.986del Chr5(GRCh37):g.33954512del p.(Thr329Lysfs*69)	NM_016180.5(SLC45A2):c.1471G>A Chr5(GRCh37):g.33944875C>T p.(Gly491Arg) (G491R)
P8	M, 7	France	No	Absent	Blue grey	White	White	Yes	Yes, esotropia	3/10 RE; 3/10 LE	NA	Grade III	NA	Grade IV	NM_016180.5(SLC45A2):c.986del Chr5(GRCh37):g.33954512del p.(Thr329Lysfs*69)	NM_016180.5(SLC45A2):c.1036G>T Chr5(GRCh37):g.33951779C>A p.(Val346Leu) (V346L)
P9	M, 16	France	No	Present, pigmented	Blue grey	White	White	Yes	No	1/10 RE; 1/10 LE	NA	Grade IV	NA	Grade IV	NM_016180.5(SLC45A2):c.986del Chr5(GRCh37):g.33954512del p.(Thr329Lysfs*69)	NM_016180.5(SLC45A2):c.986del Chr5(GRCh37):g.33954512del p.(Thr329Lysfs*69)
P10	F, 52	France	No	Absent	Blue grey	White	White	Yes	Yes, esotropia	1/10 RE; 1/10 LE	NA	Grade II	NA	Grade IV	NM_016180.5(SLC45A2):c.986del Chr5(GRCh37):g.33954512del p.(Thr329Lysfs*69)	NM_016180.5(SLC45A2):c.1166_1167del Chr5(GRCh37):g.33947470_33947471del p.(Lys389Serfs*55)
P11	F, 50	France	No	Present, amelanotic	Blue grey	White	White	Yes	No	2/10 RE; 2/10 LE	NA	Grade IV	NA	Grade IV	NM_016180.5(SLC45A2):c.986del Chr5(GRCh37):g.33954512del p.(Thr329Lysfs*69)	NM_016180.5(SLC45A2):c.1166_1167del Chr5(GRCh37):g.33947470_33947471del p.(Lys389Serfs*55)
P12	M, 65	France	No	Present, amelanotic	Blue	White	White	Yes	Yes, esotropia	2/10 RE; 2,5/10 LE	NA	Grade III	NA	Grade IV	NM_016180.5(SLC45A2):c.1273del Chr5(GRCh37):g.33947365del p.(Leu425Trpfs*9)	NM_016180.5(SLC45A2):c.1506del Chr5(GRCh37):g.33944842del p.(Thr503Profs*6)
P13	F, 62	France	No	Present, amelanotic	Blue	White	White	Yes	Yes, esotropia	1.6/10 RE; 1.6/10 LE	NA	Grade III	NA	Grade IV	NM_016180.5(SLC45A2):c.1273del Chr5(GRCh37):g.33947365del p.(Leu425Trpfs*9)	NM_016180.5(SLC45A2):c.1506del Chr5(GRCh37):g.33944842del p.(Thr503Profs*6)
P14	F, 38	France	No	Present, amelanotic	Blue	White	White	Yes	Yes, esotropia	1/10 RE; 1/10 LE	NA	Grade III	NA	Grade III	NM_016180.5(SLC45A2):c.147C>G Chr5(GRCh37):g.33984542G>C p.(Tyr49*)	NM_016180.5(SLC45A2):c.147C>G Chr5(GRCh37):g.33984542G>C p.(Tyr49*)
P15	M, 42	France	No	Present, amelanotic	Blue grey	White	White	Yes	Yes, esotropia	2/10 RE; 2/10 LE	NA	Grade III	NA	Grade IV	NM_016180.5(SLC45A2):c.986del Chr5(GRCh37):g.33954512del p.(Thr329Lysfs*69)	NM_016180.5(SLC45A2):c.1466T>C Chr5(GRCh37):g.33944880A>G p.(Leu489Pro) (L489P)
P16	F, 30	France	Yes	Present, amelanotic	Blue	White	White	Yes	No	1.6/10 RE; 2/10 LE	Hypermetropia Astigmatism	Grade IV	Grade II	Grade IV	NM_016180.5(SLC45A2):c.179T>G Chr5(GRCh37):g.33984510A>C p.(Leu60Arg) (L60R)	NM_016180.5(SLC45A2):c.179T>G Chr5(GRCh37):g.33984510A>C p.(Leu60Arg) (L60R)
P17	F, 42	France	No	Present, pigmented	Blue	White	White	Yes	Yes, esotropia	1.6/10 RE; 2/10 LE	Hypermetropia Astigmatism	Grade IV	Grade II	Grade III	NM_016180.5(SLC45A2):c.986del Chr5(GRCh37):g.33954512del p.(Thr329Lysfs*69)	NM_016180.5(SLC45A2):c.1532C>A Chr5(GRCh37):g.33944814G>T p.(Ala511Glu) (A511E)
P18	F, 66	France	Yes	Present, amelanotic	Blue	White	White	Yes	Yes, exotropia	1/10 RE; 1/10 LE	Hypermetropia Astigmatism	Grade III	Grade II	Grade III	NM_016180.5(SLC45A2):c.986del Chr5(GRCh37):g.33954512del p.(Thr329Lysfs*69)	NM_016180.5(SLC45A2):c.986del Chr5(GRCh37):g.33954512del p.(Thr329Lysfs*69)
P19	M, 2	France	No	Absent	Blue	White	White	Yes	No		NA	Grade III	Grade III	ND	NM_016180.5(SLC45A2):c.130G>A Chr5(GRCh37):g.33984559C>T p.(Gly44Arg) (G44R)	NM_016180.5(SLC45A2):c.986del Chr5(GRCh37):g.33954512del p.(Thr329Lysfs*69)
P20	F, 3	France	No	Absent	Blue	White	White	Yes	No		NA	Grade IV	NA	Grade II	NM_016180.5(SLC45A2):c.1033-6_1033-3del Chr5(GRCh37):g.33951787_33951790del p.?	NM_016180.5(SLC45A2):c.986del Chr5(GRCh37):g.33954512del p.(Thr329Lysfs*69)
**Group 2**																
P21	M, 11	France	No	Present, pigmented	Blue	Blond	Blond	Yes	Yes, exotropia	1.6/10 RE; 2/10 LE	Hypermetropia Astigmatism	NA	NA	NA	**NM_016180.5(SLC45A2):c.533_534dup** **Chr5(GRCh37):g.33982369_33982370dup** **p.(Gly179Argfs*23)**	**NM_016180.5(SLC45A2):c.977T>A** **Chr5(GRCh37):g.33954521A>T** **p.(Ile326Asn) (I326N)**
P22	M, 7	China	No	Present, pigmented	Brown	Dark blond	Brown	No	Yes, esotropia	9/10 RE; 7/10 LE	Hypermetropia	No	Grade III	Grade I	NM_016180.5(SLC45A2):c.1045G>A Chr5(GRCh37):g.33951770C>T p.(Gly349Arg) (G349R)	**NM_016180.5(SLC45A2):c.1255C>A** **Chr5(GRCh37):g.33947381G>T** **p.(Pro419Thr) (P419T)**
P23	M, 3	France	No	Present, pigmented	Blue grey	Blond	Blond	Yes	No	1/10 RE; 1/10 LE	Hypermetropia Astigmatism	Grade IV	Grade I	Grade IV	NM_016180.5(SLC45A2):c.1166_1167del Chr5(GRCh37):g.33947470_33947471del p.(Lys389Serfs*55)	NM_016180.5(SLC45A2):c.1471G>A Chr5(GRCh37):g.33944875C>T p.(Gly491Arg) (G491R)
P24	F,10	France	No	Absent	Blue	Blond	Blond	Yes	No	2/10 RE; 2/10 LE	Hypermetropia	Grade II	NA	NA	Deletion exons 1-4	NM_016180.5(SLC45A2):c.1518C>T Chr5(GRCh37):g.33944828G>A p.(Val506=)
P25	F, 50	France	No	Present, amelanotic	Blue grey	Red blond	Red blond	Yes	No	3/10 RE; 3/10 LE	NA	Grade III	NA	Grade III	NM_016180.5(SLC45A2):c.606G>C Chr5(GRCh37):g.33964078C>G p.(Trp202Cys) (W202C)	NM_016180.5(SLC45A2):c.1532C>A Chr5(GRCh37):g.33944814G>T p.(Ala511Glu) (A511E)
P26	F, 30	France	No	Present, pigmented	Blue grey	Blond	White + Blond	Yes	No	2/10 RE; 2/10 LE	Hypermetropia Astigmatism	No	Grade I	Grade IV	**NM_016180.5(SLC45A2):c.258del** **Chr5(GRCh37):g.33984433del** **p.(Val87Trpfs*26)**	NM_016180.5(SLC45A2):c.950A>G Chr5(GRCh37):g.33954548T>C p.(Tyr317Cys) (Y317C)
P27	F, 27	France	No	Present, pigmented	Blue	Blond	Blond	Yes	No	2/10 RE; 2/10 LE	HypermetropiaAstigmatism	No	Grade I	Grade IV	**NM_016180.5(SLC45A2):c.258del** **Chr5(GRCh37):g.33984433del** **p.(Val87Trpfs*26)**	NM_016180.5(SLC45A2):c.950A>G Chr5(GRCh37):g.33954548T>C p.(Tyr317Cys) (Y317C)
P28	F, 4	France	No	Absent	Blue	Blond	Blond	No	No	NA	Hypermetropia Astigmatism	No	Grade II	Grade II	NM_016180.5(SLC45A2):c.953G>A Chr5(GRCh37):g.33954545C>T p.(Arg318His) (R318H)	NM_016180.5(SLC45A2):c.1518C>T Chr5(GRCh37):g.33944828G>A p.(Val506=)
P29	F, 41	Mauritania	No	Present, pigmented	Brown	Blond	Blond	Yes	Yes microexotropia	1.2/10 RE; 1.4/10 LE	Myopia Astigmatism	Grade I	Grade III	Grade III	NM_016180.5(SLC45A2):c.1532C>T Chr5(GRCh37):g.33944814G>A p.(Ala511Val) (A511V)	NM_016180.5(SLC45A2):c.1532C>T Chr5(GRCh37):g.33944814G>A p.(Ala511Val) (A511V)
P30	M, 30	France	No	Present, pigmented	Blue	Blond	Blond	Yes	No	5/10 RE; 5/10 LE	HypermetropiaAstigmatism	Grade I	NA	NA	NM_016180.5(SLC45A2):c.806G>A Chr5(GRCh37):g.33963878C>T p.(Gly269Asp) (G269D)	NM_016180.5(SLC45A2):c.1471G>A Chr5(GRCh37):g.33944875C>T p.(Gly491Arg) (G491R)

**Patients with the severe phenotype (group 1) or mild phenotype (group 2).** Variants in bold are reported for the first time. F = female; FHP = Foveal Hypoplasia (Thomas classification); ITI = Iris Transillumination (Sjödell classification); MT = Macular Transparency (Summers classification); NA = Not Available; VA = Visual Acuity. P2 presented with a slight psychomotor developmental delay, probably related to her impaired vision. P3 presented with an association of overall delay of acquisitions and pervasive developmental disorder (stereotyping, crying, screaming, hetero- and auto aggressivity, difficult social interactions, verbal and nonverbal language disorders, restricted interests, frustration intolerance, neo-phobias) and sleep difficulties. The central nervous system MRI was normal.

## Data Availability

All data are published in the present article.

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
