# Peer review of "Oculo-Cutaneous Albinism Type 4 (OCA4): Phenotype-Genotype Correlation"

_genes, 2022, doi:10.3390/genes13122198_

Round 1

Reviewer 1 Report

Dear Authors, 

your manuscript entitled, "Oculo-Cutaneous Albinism type 4 (OCA4): phenotype-genotype correlation" Is a very interesting article trying to unveil genotype/phenotype correlation-ships in OCA4 disease.

I think that some improvements will be applied before published:

1.- In my opinion genetic results must be included within the results section. They are the main part of this manuscript to figure out genotype/phenotype correlations. Currently are in the discussion section in a chaotic disposition.

2.- Obviously, other main changes have to apply to the discussion section they include many results aspects.  A new discussion is required.

3.- I also miss including some data from other works as a table and literature review to do a better and complete table for genotype/phenotype correlations in the discussion and also in the introduction sections.

4.-Table 1 is difficult to see ( for the size of the letters and numbers)

5.-In the abstract both paragraphs start similarly please consider merging them

Author Response

Dear reviewer, dear Editor,

Please find below the point by point answer to reviewer 1

Kind Regards

Smail Hadj-Rabia

Reviewer 1

your manuscript entitled, "Oculo-Cutaneous Albinism type 4 (OCA4): phenotype-genotype correlation" Is a very interesting article trying to unveil genotype/phenotype correlation-ships in OCA4 disease. I think that some improvements will be applied before published:

1.- In my opinion genetic results must be included within the results section. They are the main part of this manuscript to figure out genotype/phenotype correlations. Currently are in the discussion section in a chaotic disposition.

We added a “Molecular findings” paragraph at the end of the results section

2.- Obviously, other main changes have to apply to the discussion section they include many results aspects.  A new discussion is required.

We completely rewrote the discussion section

3.- I also miss including some data from other works as a table and literature review to do a better and complete table for genotype/phenotype correlations in the discussion and also in the introduction sections.

Initially, we tried to build a table. However, data are lacking to build a useful table collecting clinical information.

4.-Table 1 is difficult to see ( for the size of the letters and numbers)

We may propose 2 tables: clinical manifestations (table 1) and molecular results (table 2). We preferred one table gathering all the information

5.-In the abstract both paragraphs start similarly please consider merging them

Reviewer 2 Report

The authors have ascertained 30 patients with OCA4 and attempt genotype-phenotype correlations in the collection.  20 of the patients had severe OCA similar to the OCA1 phenotype due to loss of TYR activity.  The 10 remaining patients expressed a mild form of the condition.  The severity is associated with homozygous or compound heterozygous nonsense or deletion variants of SLC45A2.  The mild phenotype was associated with at least one missense mutation.

Minor comments:

1.  In the Introduction it is stated that “Variants in 20 genes are responsible …”. The authors should cite an appropriate reference to this gene list or actually list the 20 genes in the paper (this may have already been done in the methods section for screening).

Note an additional gene has recently been suggested to cause OCA,

Al Abdi et al., Hum Genet 2022 Sept 29th

PMEL is mutated in oculocutaneous albinism

2.  In the Introduction “The encoded protein, called membrane-associated transporter protein (MATP), is involved in melanosome pH regulation and protein transport.4 “

Reference 4 is to the original report of OCA4 in 2001.  It would be better to cite a more up to date paper examining the biochemical function of SLC45A2 viz,

Liu et al. 2022, JID 142:2744-2755.e9

Ablation of Proton/Glucose Exporter SLC45A2 Enhances Melanosomal Glycolysis to Inhibit Melanin Biosynthesis and Promote Melanoma Metastasis. 

Author Response

Dear reviewer, dear Editor,

Please find below the point by point answer to reviewer 2

Kind Regards

Smail Hadj-Rabia

Reviewer 2

The authors have ascertained 30 patients with OCA4 and attempt genotype-phenotype correlations in the collection.  20 of the patients had severe OCA similar to the OCA1 phenotype due to loss of TYR activity.  The 10 remaining patients expressed a mild form of the condition.  The severity is associated with homozygous or compound heterozygous nonsense or deletion variants of SLC45A2.  The mild phenotype was associated with at least one missense mutation.

Minor comments:

  1. In the Introduction it is stated that “Variants in 20 genes are responsible …”. The authors should cite an appropriate reference to this gene list or actually list the 20 genes in the paper (this may have already been done in the methods section for screening).

The genes were listed, in italics, in the first paragraph of the method section